# Populism and health. An evaluation of the effects of right-wing populism on the COVID-19 pandemic in Brazil

Gustavo Andrey de Almeida Lopes Fernandes[1]*, Ivan Filipe de Almeida Lopes Fernandes[2]

1 Departament of Public Administration, FGV – EAESP, São Paulo, São Paulo, Brazil, 2 Centro de Engenharia, Modelagem e Ciências Sociais Aplicadas, Federal University of ABC, Santo André, São Paulo, Brazil

☉ These authors contributed equally to this work.
* gustavo.fernandes@fgv.br

**Data Availability Statement:** The data underlying the results presented in the study are available from public sources from the Brazilian

## Abstract

What are the effects of *right-wing populism* in the struggle against COVID-19? We explore data from Brazil, a country whose populist radical right-wing president was among the prominent denialists regarding the effects of the pandemic. Using cross-sectional and weekly-panel data for 5,570 municipalities during 2020, we present evidence that social distancing was weakened, and the number of cases and deaths were higher in places where the president had received greater electoral support during the 2018 presidential elections. Placebo tests using traditional right-wing vote and data on Severe Acute Respiratory Syndrome (SARS) before the pandemic outbreak indicate that the former does not correlate with health outcomes, and the populist share of the vote does not correlate with the latter. Hence, we find strong indications that right-wing populism relates to a poor response to the disease.

## Introduction

Mirandópolis is a small town in the Brazilian countryside, seven hours away from São Paulo, the largest city in Brazil. The local mayor is a loyal supporter of the Brazilian right-wing president, Jair Bolsonaro. Following his ideas, the small town never adopted social distancing during the pandemic. As a result, the city had one of the country's highest COVID-19 deaths per capita [1].

Right-wing populism has been mentioned as one of the causes hindering responses to COVID-19. Emerging literature has pointed out that populist leaders have adopted a less cautious response to the virus [2]. Anecdotal evidence indicates populism fuels the spread of the disease due to an anti-scientific approach. Countries led by populists, such as the USA, the UK, and Brazil, are among the worst performers regarding the number of cases and deaths [3].

In this paper, we analyze the Brazilian case under many factors. First, Bolsonarism is a new political movement of right-wing populism, which became the most influential political force after the 2018 presidential election. In the past, PSDB (*Partido Social Democrata Brasileiro*), a

Government. Social distancing was provided by BRASIL-IO and can be freely download at https://brasil.io/dataset/covid19/caso_full/. The authors confirm that they did not have special data access privileges and that others would be able to access the data in the same manner.

**Funding:** The author(s) received no specific funding for this work.

**Competing interests:** The authors have declared that no competing interests exist.

soft center-right party, represented right-wing views in the country; however, they were replaced by the new right-wing populist movement. By assessing the variations in right-wing voting in Brazil during the last presidential elections, we can detach the effects of traditional right-wing voting from the new populist radical right movement.

Second, recent populist experiences in Latin America were usually marked by left-wing critiques towards neoliberal economic globalization, represented in figures such as the Kirchners in Argentina, the Venezuelan Bolivarianism, and, more recently, AMLO in Mexico. In addition, it is also different from Central and East European extreme right-wing populism has xenophobic views [4–8].

Third, Brazil is a data-rich environment with more than 5570 local governments and 27 states. Besides that, within the same legal framework, there is enormous variation in support for populism among Brazilian municipal entities. Hence, the effects of populism can be analyzed regardless of the characteristics of the local political system.

Furthermore, the effects of Bolsonarism cannot be reduced to party clues, as the president was not affiliated with any party during most of the pandemic and changed party affiliation in the last year of his mandate.

Bolsonarism represents the most significant contemporary experience of radical right-wing populism in an emerging country. The emergence of evangelicals as a right-wing movement has been very recent in Brazil. During the past five years, they have evolved from a small, minority group to a large, hegemonic one. Although some churches have supported left-wing governments, a mix of very conservative beliefs has become the law. For now, they are one of the main supports of Bolsonaro's politics and policies. The president has embraced the conservative view regarding family, values, and a pro-USA view [9]. In addition, Bolsonaro's original political view has always been related to right-wing political extremism. For some authors, cruelty characterizes this kind of politics. In a democratic country, that means downplaying the effects of the disease during the pandemic or shifting the blame toward other actors [10].

There were several cases of populist leaders criticizing political globalization and global policy recommendations from a radical far-right chauvinist view in the developed world. However, none of them openly supported the herd immunity approach to combat the pandemic, ignoring efforts for rapid vaccine production [11].

Lastly, Brazil has a robust and advanced public health system, making it a particularly relevant case among emerging countries as it has more developed state health capacities than others with the same per capita income. With that being said, the Brazilian case has interesting specificities to be explored that can help shed light on some of the consequences a victory of radical right-wing populism has had in a consolidated and dynamic democracy.

## Populism and COVID-19

More than a virus, COVID-19 has affected the world in many ways: it changed how the world is connected, increased inequalities, and fostered many political changes. The pandemic also reshaped geopolitical relationships in the global system, leading to a more precarious international equilibrium [12]. The complex health response to the virus had to be complemented by various political measures. Blaming a specific group of people for the disease, or explaining it as a measure to constrain freedom, is part of a wide range of illusions seen during the pandemic [13]. Most of these illusions reduced the multidimensional reasons for the pandemic, such as climate change, global agribusiness, and very complex political process, for a more straightforward narrative. In general, something that was far from home and quickly explained to the masses [14].

Furthermore, the pandemic had multidimensional effects on national political systems. It affected voter turnout both positively and negatively, favored the rise in political relevance of extreme right-wing radical parties, as in the case of Romania, as well as undermining the political prospects of right-wing populist leaders like Donald Trump in the United States and Boris Johnson in the UK [15–18].

Political factors also play a role in response to the pandemic, as politics has been studied as one of the main factors explaining government responses to COVID-19 and health outcomes [19]. Radical right-wing parties in Europe are related to worse healthcare policies driven by "welfare chauvinism" [20]. Furthermore, lower trust in political institutions and science is essential to foster conspiracy beliefs, regardless of ideological stance [21].

When looking at the features of other types of populism, such as medical populism, there are four features: simplification, dramatization, forging divisions and invoking false knowledge claims [2]. For example, in the US, Fox News viewership negatively correlates with compliance with social isolation and underestimates COVID-19's danger [22, 23]. The literature highlights how the anti-elitist bias of populism movements and lack of trust in scientific and political institutions influences the pandemic. It favors increased suspicion of international and scientific institutions that are vital in policy responses to the sanitary crisis [24, 25].

Research has indicated the importance of political parties in dealing with COVID-19. For some, different partisan lines had an essential impact on social distancing [23, 26]. Meanwhile, others have shown that internet searches for COVID-19 and unemployment information decreased strongly among voters for Donald Trump in the 2016 presidential election [22]. Finally, some findings also claim that populism is correlated with conspiracy beliefs about COVID-19, above and beyond partisanship [27].

In the Brazilian case, however, party connections are less relevant, as the president was not a member of any party throughout the first and second waves of the pandemic. The relevant issue is the direct connection between the populist leader and his followers.

Compliance with social distancing measures in pro-government municipalities weakened after Bolsonaro initiated a vigorous campaign against social isolation measures enforced by local governments [28]. In short, the Brazilian president disseminated the idea that COVID-19 was just a "little flu" and that combating the economic crisis was more important [29].

Features of the Brazilian context highlight what some studies emphasized [30]. Understanding new populism requires analyzing how parties enter and navigate the electoral and party systems and the content of their rhetorical appeals to the public. Bolsonaro's behavior relates to a style of politics based on bad manners, which focuses on delivering performance against political correctness [31].

As extensively demonstrated in the conceptual literature on populism, even if there is an ideological aspect in its constitution, it is also necessary to understand the dichotomy between the elite and the masses as a political strategy [4, 32]. This discourse, or even logic of action, is less dependent on the leader and more associated with representational deficits [33–35].

Recent research has highlighted the correlation between the municipal share of voting for Bolsonaro and COVID-19 results [36]. However, they all relied on second-round election data and did not use placebo tests for other respiratory diseases to leverage the regression results powerfully. Without controlling for the effects of traditional right-wing votes and using first-round election data, those studies cannot separate the effects of Bolsonaro's political pressure to reduce compliance with social distancing measures from the ecological correlation between right-wing votes and worse COVID-19 results. We expect that the effects of populism could be stronger for municipalities in the first round of the elections. In the second round, electors decide more based on the rejection of the other candidate than on the voters' adherence to the chosen one's principles and values.

As the president did not affiliate with any party, theoretical party connection should be used with caution when analyzing the effects of right-wing populism in Brazil. Hence, our methodological approach separating the effects of Bolsonarism from other ideological currents in Brazilian politics is more reliable than data mining methods.

Besides party affiliation, the Brazilian case is also interesting because of its experience with the xenophobic radical right in Latin America. Even though Brazil does not have a migration problem, only a specific issue in the state of Roraima due to the Venezuelan humanitarian crisis, the immigration problem is routinely mobilized by the populist right-wing leadership [37]. The Latin American continent has been a region with a strong affinity for leftist populist governments, with the Bolivarian and Kirchnerist experiences being the most recent demonstrations [38]. The Bolsonarist experience, in turn, is more in tune with the emergence of North Atlantic populist movements that reject globalization more in its political than economic aspects, even reverberating the notions of cruelty as a political strategy [10].

In this way, Brazil shares with Central and Eastern European countries long history of racism. In Europe, racial stigma against Roma people has deep historical roots. Stigmatization remains in the collective mentality despite efforts for integration. However, Bolsonaro's vote aligns not with racial lines but with class and regional cleavages. That indicates the emergence of a new kind of right within more fluid xenophobic lines, revigorating the long Brazilian tradition of huge racial inequalities as a direct consequence of late slavery. Collective identities aid in the establishment of essential directions around whether groups are accepted or rejected. As a result, survey evidence and aggregate data eventually indicate that the core of Bolsonarism is not composed of voters among the poorest but among the middle classes and socioeconomic elites [39–41].

We understand populism as a worldview separating politics into two homogenous and antagonistic camps. On one side, the pure masses and the corrupt elite, and, on the other, as a personalistic political strategy [4]. We propose that these two facets of populism are not easily untangled. The populist leader puts himself as the best interpreter of the people's interests, values, and anxieties. In this way, the head of a populist movement will behave strategically, bypassing mediated and institutionalized mechanisms to freely communicate with his followers and promote his self-interested interpretation of the people's will. This interpretation is supposedly against the interests and values promoted by international political and scientific institutions, all biased by the elites, with the latter being a critical hotspot during a health crisis.

Mistrust of the 'mainstream' political class and growing trust in religious leaders have grown globally [16]. For instance, Hungary and Poland are cases where the extreme right took power. Meanwhile, Romania, Portugal, France, and Spain also witnessed the emergence of right-wing parties. In most countries, however, right-wing parties rise to power through parties. In Brazil, the right-wing president had no party. He also won elections before the pandemic, so his strategy to face the health emergency cannot be considered a way to rise to power. Once in charge of the presidency, he aimed to consolidate power and decrease resistance at the subnational level.

We highlight that populism should be regarded as a 'thin' ideology that, although of limited analytical use on its terms, nevertheless conveys a distinct set of ideas about the political system which interacts with the established ideational traditions of entire ideologies [4, 42].

## Data and methods

We drew on an original dataset covering Brazil's 5,570 municipalities to evaluate the connection between right-wing extremism and the spread of the pandemic. Our data represents one of the most extensive balanced datasets in the world, aligning key political aspects, economic

infrastructure, social distancing, and health capacity with deaths and cases at a local level. As we are dealing with a count date, where the unconditional mean of the outcome is much different from its variance, we estimate negative binomial models for panel and cross-sectional data [43, 44]. We control the spatial dimension using the distance between the state capital and the Brazilian federal capital and dummies for regions. Negative binomial models are adequate to handle count data as deaths and COVID-19 cases. In the S1 Appendix, as a further robustness analysis, we tested the model with cases and deaths per capita as the dependent variables, even considering that all our models control for population size.

Data of cases and deaths caused by COVID-19 were collected by BRASIL- IO(2020), which compiles all the official data produced by every Brazilian local and state government in a single database on a daily bases. Our data collection starts on February 19[th] and goes until December 31[st], 2020, a total of 45 weeks for almost all 5570 Brazilian municipalities. We employ weekly data to smooth data variation during business days and weekends. All municipalities enter the sample on February 19[th]. In total, we have a balanced municipal-week dataset. Our time frame concerns only 2020 since a massive vaccination campaign started in January 2021, despite the difficulties created by the federal government for a broad vaccination of the Brazilian population. Consequently, the Covid response involves more strategies than NPIs. Future studies should analyze the implications.

We use the vote for Bolsonaro in the 2018 first-round elections in each Brazilian municipality as a proxy for commitment to right-wing populism. We test its relation with COVID-19 responses using cross-section and panel data models on Brazilian municipalities. The local share of votes for the most competitive right-wing candidate during the 2014 elections is used as a placebo test that allows us to separate the effect of the presidency's interventions against NPI measures from the potential effects of the Brazilian political right itself.

Geo-located data from cellphones by municipalities measure social isolation compliance by Inloco[®]. We present details about data collection in the S1 Appendix.

As robustness checks for the effect of populism, we tested the impact of the moderate right-wing vote, using data from the 2014 election and data for other severe respiratory diseases in 2018 as a placebo test for overall respiratory diseases unrelated to COVID-19. Data on Severe Acute Respiratory Syndrome (SARS) was initially collected in Brazil as a response to the Zika virus pandemic. We collected data on SARS from 2018 in all Brazilian municipalities on the Health Ministry database. We assume that there are no changes concerning SARS transmission in Brazil between 2018 and 2020 regarding socioeconomic and epidemiological variables, except for the political context and the surge of COVID-19, which makes it a valuable placebo.

Finally, we used the moderator-mediator variable distinction approach and other methods of exploring causality to explore the possible mechanism: the populist effect on social isolation compliance [45, 46].

Lastly, we obtained control variables from the Brazilian Bureau of Statistics and the Ministry of Education. Table 1 shows descriptive statistics. Fig 1 depicts the distribution of Covid cases and deaths in Brazil. It also shows the cases of respiratory diseases in 2018. Fig 2 presents the distribution of the share of right-wing vote in the country in the 2014 and 2018 elections.

We show details on variables and complete results in the S1 Appendix.

## Results

Table 2 shows **panel data estimates** for weekly deaths and confirmed cases of infection. Bolsonaro's local share of votes is statistically significant in all models. The larger the share of votes received, the more cases and deaths are seen. Columns (1) to (4) test the effect on weekly cases. The last four columns consider the impact on weekly deaths. Results

**Table 1. Descriptive statistics.**

| Variable | Obs. | Mean | Std. Dev. | Min | Max |
|---|---|---|---|---|---|
| **Weekly Covid Cases** | 5,570 | 63.545 | 345.852 | 0 | 16607 |
| **Weekly Deaths due to Covid** | 5,570 | 1.213 | 9.299 | 0 | 480 |
| **Confirmed cases in the year** | 5,570 | 1367.03 | 7916.473 | 2 | 401718 |
| **All Deaths due to Covid in the year** | 5,570 | 34.912 | 324.955 | 0 | 15679 |
| **Cases of Respiratory Diseases in 2018** | 5,570 | 8.701 | 97.102 | 0 | 4227 |
| **Bolsonaro's Share of the Vote in 2018** | 5,570 | 38.726 | 18.982 | 1.941 | 83.893 |
| **Right Wing Share of the Vote in 2014** | 5,570 | 32.806 | 17.34 | 1.514 | 82.562 |
| **Social Isolation—Annual Average** | 4,778 | 39.636 | 3.322 | 24 | 56.604 |
| **Hospital Beds** | 5,570 | 53.872 | 320.064 | 0 | 14822 |
| **Additional Hospital Beds** | 5,570 | 5.459 | 51.275 | 0 | 2586 |
| **Log Local Wealth** | 5,570 | 12.309 | 1.405 | 9.472 | 20.366 |
| **Log Population** | 5,570 | 2.568 | 1.172 | -0.208 | 9.402 |
| **Share Pop. Older than 60 years** | 5,565 | 0.176 | .05 | 0.034 | .435 |
| **Log Distance to Federal Capital** | 5,507 | 6.871 | .525 | 0 | 7.961 |
| **Log Distance to State Capital** | 5,507 | 5.258 | .878 | 0 | 7.297 |
| **Child Mortality Rate per 1.000** | 5,570 | 12.889 | 12.821 | 0 | 181.820 |
| **Life Expectancy** | 5,570 | 66.982 | 0.509 | 54.35 | 78,10 |
| **Education Index of Vulnerability** | 5,547 | 4.726 | 0.509 | 3.514 | 5.952 |

show that the traditional right-wing vote (2014) correlates with fewer weekly cases and deaths.

On the other hand, Bolsonaro's share of the vote (2018 vote) correlates with more cases and deaths. Furthermore, effects are more potent when the traditional right-wing electorate is analyzed independently from Bolsonaro's share of the vote. Incidence rate ratio results indicate that a 1-point increase in Bolsonaro's share of the vote increases the municipal rate of cases by a factor of 1.004 and deaths by a factor of 1.008 while holding all other variables constant (columns (4) and (8)). Although important, these numbers are smaller than the figures found by some studies for socioeconomic status [47]. The findings are the same when we use cases and deaths of Covid per 1000 inhabitant as the dependent variable. Table 4A in the S1 Appendix presents the results.

Table 3 shows **cross-sectional findings** concerning all respiratory disease cases during 2018 and the total number of cases and deaths from COVID-19 in 2020. For the former, we

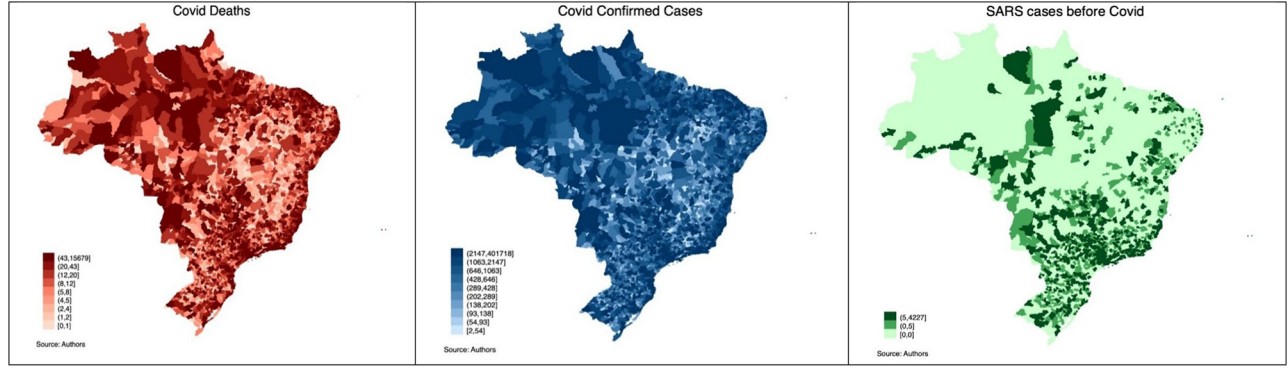

**Fig 1. Respiratory diseases in Brazil.**

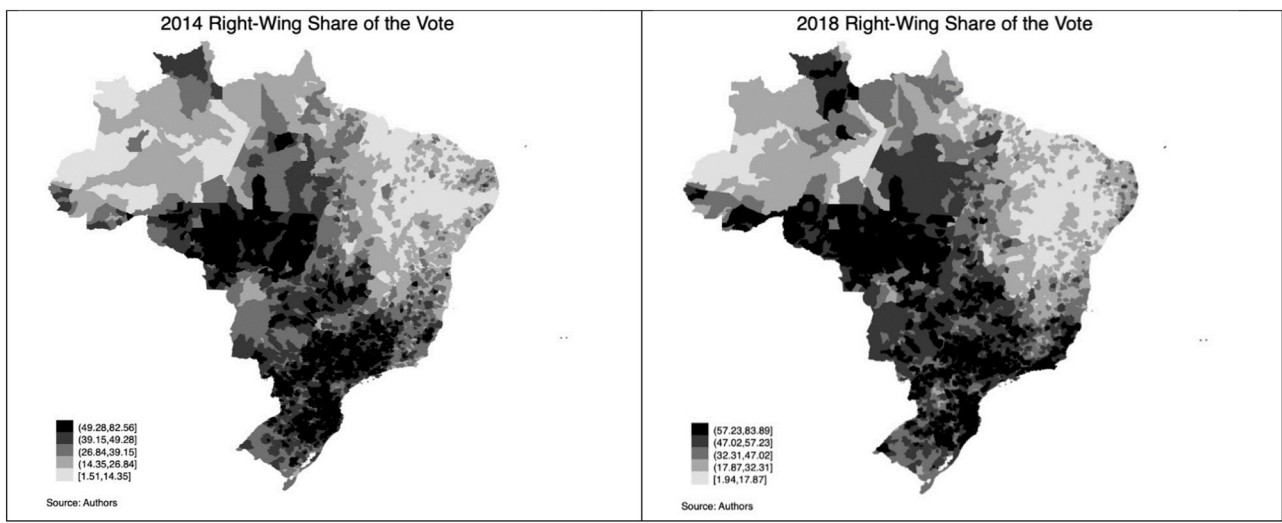

**Fig 2. The right-wing vote in Brazil.**

aggregated all cases in 2018. Findings show that Bolsonaro's vote share is not statistically significant when we include controls. That is an important result since any omitted variable that could explain the main results would be seen here. However, when it gets to COVID-19 outcomes, it has a statistically significant impact, as shown in Table 1, and estimates are larger—see columns (4) and (5). The results are the same when we use cases of SARS per 1000 inhabitants, Covid per 1000 inhabitant cases, and Covid per 1000 inhabitant deaths as the dependent variables. Table 4B in the S1 Appendix presents the results.

**Table 2. Panel data analysis.**

| | (1) | (2) | (3) | (4) | (5) | (6) | (7) | (8) |
|---|---|---|---|---|---|---|---|---|
| | Weekly-Cases | | | | Weekly-Deaths | | | |
| **Bolsonaro's Share of the Votes 2018** | 0.002 [0.000]*** | 0.012 [0.000]*** | 0.005 [0.001]*** | 0.005 [0.001]*** | 0.007 [0.001]*** | 0.023 [0.001]*** | 0.012 [0.001]*** | 0.008 [0.001]*** |
| **Right-Wing-Share of the Vote 2014** | | -0.013 [0.000]*** | -0.003 [0.001]*** | -0.005 [0.001]*** | | -0.022 [0.001]*** | -0.007 [0.001]*** | -0.010 [0.001]*** |
| **Weekly-Social-Isolation Average one-lag** | | | | -0.025 [0.001]*** | | | | -0.031 [0.002]*** |
| **Weekly-Social-Isolation Average two-lags** | | | | 0.032 [0.001]*** | | | | 0.038 [0.002]*** |
| **Controls** | | | x | x | | | x | x |
| **Municipalities** | 5,570 | 5,570 | 5,486 | 4,680 | 5,570 | 5,570 | 5,486 | 4,680 |
| **N** | 245,080 | 245,080 | 241,384 | 164,967 | 245,080 | 245,080 | 241,384 | 164,967 |

Negative binomial regression coefficients with standard errors in brackets

***$p< 0.001$,

**$p< 0.01$,

*$p< 0.05$,

+$p<0.1$, respectively.

Controls: Hospital Beds, Additional Hospital Beds in 2020, Log of the size of the local economy, Log Population, Share Pop. Older than 60 years, Distance to Federal Capital, Distance to State Capital, Child Mortality Rate, Life Expectancy in 2000, and Educational Vulnerability. We employed state dummies in (3) and regional dummies for (4) and (7).

**Table 3. Cross-sectional analysis.**

|  | (1) | (2) | (3) | (4) | (5) | (6) |
|---|---|---|---|---|---|---|
|  | Severe Acute Respiratory Syndrome (SARS) | | | | Covid-2020 Cases | Covid-2020 Deaths |
|  | 2018-Cases | | | | | |
| **Bolsonaro's Share of the Votes 2018** | 0.082 [0.014]*** | 0.141 [0.019]*** | 0.007 [0.015] | 0.008 [0.019] | 0.019 [0.005]*** | 0.024 [0.005]*** |
| **Right-Wing-Share of the Vote 2014** |  | -0.071 [0.020]*** | 0.026 [0.013]* | 0.003 [0.011] | -0.011 [0.005]* | -0.013 [0.004]*** |
| **Controls-I** |  |  | x | x | X | x |
| **Controls-II** |  |  |  | x | X | x |
| **N** | 5,570 | 5,570 | 5,570 | 5,486 | 5,486 | 5,486 |

Negative binomial regression coefficients with standard errors in brackets.

***$p < 0.001$,

**$p < 0.01$,

*$p < 0.05$,

+$p < 0.1$, respectively.

SEs clustered by state. Bivariate relations for columns 1 and 2.

Controls-I: Log of the size of the local economy and Log Population

Controls-II: Hospital Beds, Additional Hospital Beds in 2020, Share Pop. Older than 60 years, Distance to Federal and State Capital, Child Mortality Rate, Life Expectancy in 2000, and Educational Vulnerability.

Table 4 explores **the effect of support for Bolsonaro on social distancing** as a likely mechanism. Results indicate that the populist right-wing vote is associated with lower compliance with social isolation, whereas the traditional right-wing does not. We estimated that a 10.0 percentage-point increase in Bolsonaro's vote share decreases social isolation by, on average, 2.5 percentage points. Other studies found 11,50% to 15,20% for an 80% increase in the republican share of the vote [26].

Mediation analysis presented at the bottom level of Table 4 indicates that social isolation mediates the effects of Bolsonaro's share of votes on deaths. We estimated that 21.8% of the effects of Bolsonaro's votes on deaths are mediated by the decline in social isolation compliance. Complete analysis in the S1 Appendix.

Our findings suggest that the president's electoral base has been associated with less social isolation, jeopardizing policies for mitigation made by state and local Brazilian authorities and worsening the pace of infection and consequent COVID-19 deaths. The effects on social isolation compliance are a likely mechanism but not the only way Bolsonarism led to a worse COVID-19 response. Lastly, we must point out that we tested our estimations and placebos using observational data.

## Discussion

The analysis of the Brazilian case offers a different blend of populism and nationalism [30, 42, 47]. Despite not being a political party member, extensive literature shows that Bolsonaro is undeniably an extreme right-wing leader. We expect that some of the Brazilian population has conservative views; however, the views that Bolsonaro carries are extremist. As a result, conservative views and populism are essential to understanding the disease's dynamic. Nevertheless, it is also true that the type of views shared (i.e., antiscience) by him and the role Bolsonaro played as president (i.e., blaming the left) makes right-wing politics crucial to what happened in Brazil.

However, Bolsonaro's nationalism is not against a specific ethnic group such as the Roma in Eastern Europe or another country such as China or Trump's speech. It is against an old

**Table 4. Right-wing populism and social isolation.**

| | (1) | (2) | (3) | | (4) | (5) | (6) |
|---|---|---|---|---|---|---|---|
| | OLS: Panel-Data analysis | | | | OLS: Cross-section analysis | | |
| Bolsonaro's Share of the Votes 2018 | -0.069 [0.007]*** | | -0.026 [0.010]* | | -0.070 [0.008]*** | | -0.026 [0.011]* |
| Right-Wing-Share of the Vote 2014 | | -0.052 [0.007]*** | -0.014 [0.011] | | | -0.051 [0.007]*** | -0.013 [0.012] |
| Municipalities | 4,778 | 4,778 | 4,760 | | | | |
| Controls | | | X | | | | X |
| N | 173,742 | 173,742 | 172,846 | | 4,778 | 4,778 | 4,747 |

| **Mediation Analysis of Social Isolation Compliance—Accumulated deaths per capita (one-hundred-thousand habitants)** | | | | | | | |
|---|---|---|---|---|---|---|---|
| | OLS: Panel-Data log(deaths per weak) | | | Bayesian Mechanism Analysis | | | |
| | (7) | (8) | | | | | |
| Bolsonaro's Share of the Votes in 2018 | 0.004 [0.001***] | 0.003 [0.001**] | | *Estimate* | *Lower-95% CI* | *Upper-95% CI* | *p-value* |
| Right-Wing-Share of the Vote in 2014 | -0.006 [0.001***] | -0.007 [0.001***] | ACME | 0.00067 | 0.0007 | 0.001 | <0.000 |
| | | | ADE | 0.00251 | 0.0002 | 0.001 | 0.02 |
| Weekly-Social-Isolation Average one-lag | | -0.031 [0.002***] | Total Effect | 0.00318 | 0.0008 | 0.010 | <0.000 |
| | | | Prop. Mediated | 21.79% | 7.29% | 50.00% | <0.000 |
| Controls+Week and State-Dummies | X | X | | | | | |
| Municipalities | 4,120 | 4,120 | | 4,760 | | | |
| N | 171,027 | 171,027 | | 172,846 | | | |

Linear regression coefficients with standard errors in brackets

***p< 0.001,

**p< 0.01,

*p< 0.05, respectively.

Week stands for Epidemiological Week. All models are controlled by state dummies. SEs clustered by state.

Controls I and II

ACME: Average Causal Mediated Effects. ADE: Average Direct Effects. Prop. Mediated: proportion of the mediated effect

communist threat that comes from abroad. In some ways, Brazilian populism generates an invisible enemy. As it is not in a specific place or group of people, it can be everywhere. That omnipresent enemy empowers a thin ideology that characterizes Bolsonaro's right-wing views [42].

Besides, the Brazilian case strongly highlighted the effect of populism in countries where the party system lacks institutionalization and where authorities in the public sector usually show low-performance levels—a case like those of the CEE and the Global South [33].

In some middle-income federal governments, the central government acted politically by transferring responsibility to subnational governments. Consequently, health policies lacked adequate coordination, resulting in poor performance. It also prevented policies that motivated people to adopt the best behavior toward the disease [48, 49]. Brazilian policies were the least stringent across the Americas. There was also wide variation in national-level NPI responses to the COVID-19 pandemic. Ultimately, states led by oppositional parties took the more stringent policies [50]. However, these measures were not enough. The disease spread unmitigated for the whole country, reaching local catastrophes such as in Manaus [51].

We see the Brazilian case as an example of Punt Politics when national governments defer or deflect—de jure or de facto—responsibility to sub-national entities for crucial decisions that require centralized stewardship—in Brazil driven by the presidency. That resulted in fragmented, uncoordinated responses at odds with health needs and consistency with evidence.

Results indicate that better public policy response demands more accountability than was seen. Left or right-wing dictatorships have appeared all around the world. Radical views have structured some of them. Most of them used to blame the different types of enemies: immigrants, different ethnic groups, and, in the Brazilian case, an international communist threat. Clear political responsibilities can decrease the room for populist views because it would reduce how populists can blame others for bad public policy results. That is an essential lesson for the international community, regardless of the level of development.

Finally, for emerging countries, such as Latin America, more structured long-term social policies could also reduce the space populist politicians use to manipulate people. NPI measures could have been more straightforwardly implemented if they were associated with social policies reducing the economic downside. As people were encouraged to boycott social distancing, which was blamed for being the cause of the economic downturn, the disease spread unprecedentedly. A different story could have been written if the President had told people that staying at home would protect lives. Besides, in the end, it would save the economy.

The Brazilian experience shows how the lack of adequate accountability and the surge of radical views have produced unparalleled health and economic disaster.

## Limitations of the study

Our analysis relies on ecological data. We have the leverage to estimate the effect of right-wing populism on COVID-19 results due to denialism and political pressures from the Brazilian president. However, further research should test the micro-foundations of our hypothesis using individual-level data. A preliminary analysis of Brazilian data corroborates our findings [52].

Although our placebo strategy helps shed light on the relationship between COVID-19 results and political variables, our estimation does not rely on exogenous variation to estimate Bolsonarism's effect on deaths and cases. Hence, we are confident that we can capture the structure of the relations. However, more research should be done to estimate the actual effects of Bolsonaro's politics on the number of cases and deaths caused by COVID-19 in Brazil.

A further line of inquiry is to identify how socioeconomic status modulates the effect of Bolsonarism, as the right-wing populist movement in Brazil is associated with middle- and high-income individuals.

## Conclusion

Our paper analyzes the relationship between support for Bolsonaro and responses to COVID-19. Findings show that the impact of extreme right-wing views on the pandemic has been striking in Brazil. We present evidence that in the places where the president had more electoral support, the impact of COVID-19 was worse. The effects on social isolation compliance are a likely mechanism but not the only one. Campaigns against mask usage and quarantine policies can be other likely potential mechanisms.

Right-wing populism has resulted in severe social damage and, for some Brazilians, a fatal outcome. As unobservable omitted variable bias is a ubiquitous problem in political science conclusions obtained from observational data, our estimations should be considered with caution.

Future studies should analyze how populism affects the policies adopted to stimulate the economy, social isolation by subnational governors, and the early preparation of the health system. Lastly, an important research topic on the effects of populism on the pandemic is related to the politics of vaccination, whose anecdotal evidence points to being another broad field of action for populist leaders [53, 54].

## Supporting information

**S1 Appendix.**
(DOCX)

## Acknowledgments

We thank Gabriela Pinheiro for her valuable research assistance.

## Author Contributions

**Conceptualization:** Gustavo Andrey de Almeida Lopes Fernandes.

**Investigation:** Gustavo Andrey de Almeida Lopes Fernandes, Ivan Filipe de Almeida Lopes Fernandes.

**Methodology:** Gustavo Andrey de Almeida Lopes Fernandes, Ivan Filipe de Almeida Lopes Fernandes.

**Project administration:** Gustavo Andrey de Almeida Lopes Fernandes.

**Writing – original draft:** Gustavo Andrey de Almeida Lopes Fernandes, Ivan Filipe de Almeida Lopes Fernandes.

**Writing – review & editing:** Gustavo Andrey de Almeida Lopes Fernandes, Ivan Filipe de Almeida Lopes Fernandes.

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
