## [Decision Letter · Decision Letter 0]

10 Aug 2022

PONE-D-22-14260Populism and health. An evaluation of the effects of right-wing populism in the COVID-19 pandemic in BrazilPLOS ONE

Dear Dr. Fernandes,

Thank you for submitting your manuscript to PLOS ONE. After careful consideration, we feel that it has merit but does not fully meet PLOS ONE’s publication criteria as it currently stands. Therefore, we invite you to submit a revised version of the manuscript that addresses the points raised during the review process.

ACADEMIC EDITOR: the theme of the article is relevant, but there are already published articles showing that there were more cases of COVID-19 and higher mortality in Brazilian municipalities where JB had more votes. The article by Xavier et al. (2022), published in The Lancet Regional Health-Americas, was widely commented on in the media. Cabral et al. (2021) and Constantino et al. (2021) published other articles on this topic. Surprisingly, none of these previous studies were cited in the present article. Thus, for the present article to be considered for publication, the authors need to describe this previous research and be convincing about what the present article can add to the findings of these authors. Without this effort, the present article will be seen as "more of the same", and the conclusions that JB had more votes where there were more cases and deaths from COVID-19 are already known.

We look forward to receiving your revised manuscript.

Kind regards,

Diego Augusto Santos Silva, Ph.D.

Academic Editor

PLOS ONE

Journal Requirements:

Reviewers' comments:

Reviewer's Responses to Questions

**Comments to the Author**

1. Is the manuscript technically sound, and do the data support the conclusions?

Reviewer #1: Partly

Reviewer #2: Partly

Reviewer #3: Yes

2. Has the statistical analysis been performed appropriately and rigorously? 

Reviewer #1: Yes

Reviewer #2: No

Reviewer #3: Yes

3. Have the authors made all data underlying the findings in their manuscript fully available?

Reviewer #1: Yes

Reviewer #2: No

Reviewer #3: Yes

4. Is the manuscript presented in an intelligible fashion and written in standard English?

Reviewer #1: Yes

Reviewer #2: Yes

Reviewer #3: Yes

5. Review Comments to the Author

Reviewer #1: This is a timely paper which has to be taken into consideration for publication in PLOS ONE. Authors have approached a new line of investigation in populism studies, meaning what are the effects of right-wing populism in the struggle against COVID-19 in Brasil. The findings of the paper highlight that right-wing populism is an indicator for a poor response to the COVID-19 disease.

Besides the merits of the paper which are obvious, after re-reading attentively this paper I think there are several issues on which the authors have to think to revise or to improve throughout the paper.

1) In the introduction it is needed to present how this paper pushes forward the current knowledge in populism studies or what it brings new at international level of populism and coronavirus studies. So, the question that should be raised is how this study is positioned in the current broader CEE contexts or why the Brasilian case in this paper is so different from other cases we already know. These ideas can be then further connected in the last part of the paper and see how the findings of this paper differ to the previous ones in contexts of populism and coronavirus in different other countries.

The theoretical background of the paper has to be a little bit improved with some recent studies on coronavirus and on the emergence of the far-right studies in different countries worldwide. First, I think some works on COVID-19 should be mentioned. There are several special issues launched at different journals on COVID-19. For instance the case of transnational migrant women in India is important, also the case of transnational workers from Romania who tried hard to work in western Europe (see a study in Eurasian Geography and Economics, 2020). Second, with connection to far-right populism there is the case of the AUR party in Romania – see Vesalon L. and Popescu L. in journal East European Politics (2022) and see Doiciar Claudia et al (2021) in Geographica Pannonica. The first paper showed how the message of AUR is similar to neo-marxists, while the second highlights how the far-right/new nationalists from AUR capitalize sensitive environmental issues in order to gain votes. Several studies on far-right in other countries of Europe should be also mentioned (see Austria, Hungary, France who have far-right parties and they were vocal during the coronavirus times, especially in connection to wearing masks, anti-vaccination protests etc).

Moreover, I think that several examples of how the poor people are much easy to be influenced by the far-right discourse, because these people are more vulnerable. For instance, in Europe it is the case of the Roma people who are used by populists and far-right leaders as an electoral basin in order to win elections (see the study by Mereine Berki B. et al, 2020 in Geographica Pannonica, on how stigma against the poor Roma in Hungary can appear basing among others on historical and political manipulation of these people). The same is highlighted in an East-Central European border rural area (Romanian-Serbian) for the Roma in Romania (see a study of Covaci and Jucu, 2021 in journal Identities).

2) Methods and data interpretation are strong. Just to highlight in 1-2 sentences the limitations of the data and methods.

3) Because there is not a clear-cut discussion section in this paper, I would like to see (before the conclusions section) about 1-2 paragraphs connecting better the results of this study with the literature review on coronavirus and far-right populism. Also, some policy recommendations can be shortly addressed.

4) Conclusions should be a bit expanded by better showing the international, regional (Latin American) and Brasilian implications of this study.

5) The reference list is short, it is made up of about 20 references. I think the reference list should be made of about 50 sources. This aspect is in connection to my above point 1) where I suggested some references, but besides those examples authors have to add more references on extremism/nationalism/far-right populists in Europe, Latin America, Asia etc.

Reviewer #2: [1] As a first comment, we can consider that the theme of the article is relevant, but there are already published articles showing that there were more cases of COVID-19 and higher mortality in Brazilian municipalities where JB had more votes. The article by Xavier et al. (2022), published in The Lancet Regional Health-Americas, was widely commented on in the media. Cabral et al. (2021) and Constantino et al. (2021) published other articles on this topic. Surprisingly, none of these previous studies were cited in the present article. Thus, for the present article to be considered for publication, the authors need to describe this previous research and be convincing about what the present article can add to the findings of these authors. Without this effort, the present article will be seen as "more of the same", and the conclusions that JB had more votes where there were more cases and deaths from COVID-19 are already known.

Xavier, D. R., e Silva, E. L., Lara, F. A., e Silva, G. R., Oliveira, M. F., Gurgel, H., & Barcellos, C. (2022). Involvement of political and socio-economic factors in the spatial and temporal dynamics of COVID-19 outcomes in Brazil: A population-based study. The Lancet Regional Health-Americas, 100221.

Cabral, S., Ito, N., & Pongeluppe, L. (2021). The disastrous effects of leaders in denial: evidence from the COVID-19 crisis in Brazil. Available at SSRN 3836147.

Constantino, S. M., Cooperman, A. D., & Moreira, T. M. (2021). Voting in a global pandemic: Assessing dueling influences of Covid‐19 on turnout. Social science quarterly, 102(5), 2210-2235.

[2] Some editing for English language is required throughout the manuscript due to too many mistakes. These are just a few examples found in the abstract:

Change “Placebo tests indicates that” to “Placebo tests indicate that”.

Change "indicator for a poor response" to "indicator of a poor response".

Other weird sentences and word choices need revision.

[3] The abstract should be self-explanatory, that is, the reader should be able to grasp the subject without referring to the main text. In this sense, it is not clear what "placebo tests" means in the abstract.

[4] Abstract, third line. I would suggest to use “effects of the disease” or “effects of the COVID-19” rather than “effects of SARS-CoV-2”. SARS-CoV-2 refers to the virus, while COVID-19 refers to the disease that it causes in humans.

[5] Introduction. It is not clear how important this sentence is in the context of the introduction: “In the past, right-wing views were represented by PSDB (Partido Social Democrata Brasileiro), a soft right-center party”. Perhaps the PSDB was closer to Tony Blair's "Third Way" vision for Europe than a far-right party, but I believe that this information is not important in the introduction of the article.

[6] Introduction. The authors state that “… the effects of Bolsonarism cannot be reduced to party clues, as the president was not affiliated to any party during most part of the pandemics and changed party affiliation in the last year of his mandate”. Understanding the support that JB has from the Brazilian population requires knowledge of many factors, including religious, cultural, and social issues. Brazilian society is patriarchal and conservative, and the support for JB's ideas goes far beyond party-political issues. I believe that these articles can help the authors better structure the introduction of the article by describing the social structure of the country during the pandemic period:

Farias, D. B. L., Casarões, G., & Magalhães, D. (2022). Radical right populism and the politics of cruelty: The case of COVID-19 in Brazil under President Bolsonaro. Global Studies Quarterly, 2(2), ksab048.

Burity, J. (2021). The Brazilian conservative wave, the Bolsonaro administration, and religious actors. Brazilian Political Science Review, 15.

Barberia, L. G., & Gómez, E. J. (2020). Political and institutional perils of Brazil's COVID-19 crisis. Lancet (London, England), 396(10248), 367.

[7] Introduction. “In short, the Brazilian president disseminated the idea that COVID-19 was just a “little flu” and combating the economic crisis was more important”. I would suggest to cite this article:

Dyer, O. (2020). Covid-19: Bolsonaro under fire as Brazil hides figures. BMJ, 369, m2296.

[8] Introduction. In summary, the introduction section needs to be extensively reformulated in an effort to present previous articles on the topic, possible "gaps" in this research, and essential information for the reader to understand the context of the JB government's political landscape of misinformation and omission.

[9] The methods section does not make clear how respiratory disease data were included in the statistical model to test for a "placebo effect." The methods section should contain enough information to ensure that readers can reproduce the statistical analyses.

[10] In the conclusion section, the authors argue that the "Findings show that the impact of extreme right-wing views on the pandemic has been striking in Brazil". However, in the introduction of the article, the authors discuss that "In the Brazilian case, however, party connections are less relevant, as the president was not a member of any party throughout the first and second wave of the pandemic". I think that these sentences are contradictory in some ways. It does not appear that right-wing views are related to the number of COVID-19 cases or deaths, but these relationships are determined by a wide range of issues, including traditional, conservative, religious views, and even a sense of Brazilian Mccarthyism that demonizes perceived "left-wing" social actions. So the problem does not seem to be that JB is a right-wing politician, but what causes greater numbers of the disease is his populism and conservative views of the population.

[11] I have doubt that the statistical methods used by the authors are the most appropriate for the data. The authors do not comment on model assumptions, such as the linearity of the relationship between independent and dependent variables, and the diagnosis of homoscedasticity and distribution of residuals. The model does not include an offset variable or a population size weighting strategy. The model also does not include spatial structures that would allow testing on maps for clusters where JB has more votes and COVID-19 cases were more frequent. The previous papers by Xavier et al. (2022), Cabral et al. (2021), and Constantino et al. (2021) show maps and graphs, which are more understandable than the results shown in the present study.

[12] Potential limitations of the study are not presented.

[13] The discussion section is very shallow and it needs major modification.

[14] At the end of the article, the sentence beginning with “Lastly, an important research topic on the effects of populism…” needs some revision, and this argument could be supported by the following articles:

Daniels, J. P. (2021). Health experts slam Bolsonaro's vaccine comments. The Lancet, 397(10272), 361.

Boschiero, M. N., Palamim, C. V. C., & Marson, F. A. L. (2021). COVID-19 vaccination on Brazil and the crocodile side-effect. Ethics, medicine, and public health, 17, 100654.

Reviewer #3: This paper is very well-written and provides an excellent combination of theory on populism and evidence surrounding subnational responses to the covid-19 pandemic in Brazil.

I have several recommendations that I think will improve the overall contribution, which I already find compelling. First, I suggest adding a “limitations” section along the lines of what we’ve seen recently in many public health and medical journal publications surrounding covid-19. Most of these limitations are methodological, which I think is completely acceptable given the real-time data collection and analysis in question.

causal identification is very difficult in this context, which I think is reasonable and currently common in the top medical journals for work on covid. However, I suggest moderating some of the claims in the paper and describing why the models presented represent the best-case scenario in the current climate and how the results can still help build theory and inform government practice. Another limitation is simply the lack of municipal level covariates available and/or the way that aggregating to the municipal level obscures unobserved and (probably) unobservable submunicipal variation.

Next, I suggest defending the case selection a little more thoroughly. For example, are the results from Brazil generalizable beyond the country? Why/why not? I agree with the authors’ rationale and I think they should make an even larger claim: that brazil, due to data availability and municipal variation, is the only country where they could plausibly test hypotheses against such broad, deep data.

Defending the particular timeframe under consideration should also be part of the next revision. I see lots of reasons to focus on the timeframe under consideration in the article, but I would like to see those reasons articulated thoroughly. For example, because the covid response shifted from NPIs to vaccines after 2020.

Finally, there is recent literature on Brazil and Mexico and on subnational covid issues in Latin America in general that should be included in the review and with which this article can enter into conversation:

Knaul, F. M., Touchton, M., Arreola-Ornelas, H., Atun, R., Anyosa, R. J. C., Frenk, J., ... & Victora, C. (2021). Punt politics as failure of health system stewardship: evidence from the COVID-19 pandemic response in Brazil and Mexico. The Lancet Regional Health-Americas, 4, 100086.

Touchton, Michael, Felicia Marie Knaul, Héctor Arreola-Ornelas, Thalia Porteny, Mariano Sánchez, Oscar Méndez, Marco Faganello et al. "A partisan pandemic: state government public health policies to combat COVID-19 in Brazil." BMJ global health 6, no. 6 (2021): e005223.

Knaul, F. M., Touchton, M. M., Arreola-Ornelas, H., Calderon-Anyosa, R., Otero-Bahamón, S., Hummel, C., ... & Sanchez-Talanquer, M. (2022). Strengthening Health Systems To Face Pandemics: Subnational Policy Responses To COVID-19 In Latin America: Study examines policy responses to COVID-19 in Latin America. Health Affairs, 41(3), 454-462.

Testa, Paul F., Richard Snyder, Eva Rios, Eduardo Moncada, Agustina Giraudy, and Cyril Bennouna. "Who Stays at Home? The Politics of Social Distancing in Brazil, Mexico, and the United States during the COVID-19 Pandemic." Journal of Health Politics, Policy and Law (2021).

Castro, Marcia C., Sun Kim, Lorena Barberia, Ana Freitas Ribeiro, Susie Gurzenda, Karina Braga Ribeiro, Erin Abbott, Jeffrey Blossom, Beatriz Rache, and Burton H. Singer. "Spatiotemporal pattern of COVID-19 spread in Brazil." Science 372, no. 6544 (2021): 821-826.

Knaul, Felicia, Héctor Arreola-Ornelas, Thalia Porteny, Michael Touchton, Mariano Sánchez-Talanquer, Óscar Méndez, Salomón Chertorivski et al. "Not far enough: Public health policies to combat COVID-19 in Mexico’s states." Plos one 16, no. 6 (2021): e0251722.

The rest of the paper is very well-done. By engaging with the recent literature on the speicifc subject, the paper will be easier to find for scholars and policymakers working in the area.

6. PLOS authors have the option to publish the peer review history of their article (what does this mean?). If published, this will include your full peer review and any attached files.

Reviewer #1: No

Reviewer #2: No

Reviewer #3: No

---

## [Author Response · Author response to Decision Letter 0]

20 Oct 2022

Dear Prof. Professor Diego Augusto Santos Silva,

We want to thank you and the three reviewers for your thoughtful comments. We endeavored to incorporate the changes you suggested, and we believe our manuscript improved as a result. Below, we intersperse our reactions and revisions to your suggestions into a verbatim full text of the manuscript reviews. We adopt the following conventions:

 Full verbatim quotes of the editor’s decision letter and reviewers’ reports are between commas (“ ”), which have been lightly edited for clarifications or to conserve space. 

 Our responses to the editor’s and reviewers’ suggestions are in standard text.

 The specific locations in the manuscript and appendix where incorporated revisions are in UPPERCASE LETTERS, as well as we repeat the changes in this letter in bold. 

 Academic Editor

“The theme of the article is relevant, but there are already published articles showing that there were more cases of COVID-19 and higher mortality in Brazilian municipalities where JB had more votes. The article by Xavier et al. (2022), published in The Lancet Regional Health-Americas, was widely commented on in the media. Cabral et al. (2021) and Constantino et al. (2021) published other articles on this topic. Surprisingly, none of these previous studies were cited in the present article. Thus, for the present article to be considered for publication, the authors need to describe this previous research and be convincing about what the present article can add to the findings of these authors. Without this effort, the present article will be seen as "more of the same", and the conclusions that JB had more votes where there were more cases and deaths from COVID-19 are already known”.

RESPONSE: We thank the Reviewer and Editor for their indication and incorporate in our analysis the results of Xavier et al. (2022) and Constantino et al. (2021) [POPULISM AND COVID-19].

Despite the indication of the text by Cabral et al. (2021), we chose not to incorporate it in our discussion because it is not yet published, which would violate the journal's recommendation in https://journals.plos.org/plosone/s/submission-guidelines#loc-references

[Do not cite the following sources in the reference list: Unavailable and unpublished work, including manuscripts that have been submitted but not yet accepted (e.g., “unpublished work,” “data not shown”). Instead, include those data as supplementary material or deposit the data in a publicly available database.]

Besides, we incorporate the paper by Constantino et al. (2021). in another section because it is more directly related to the electoral consequences of COVID-19 in the 2020 Brazilian municipal election than to the analysis of the mediating effects of policy on pandemic outcomes, which is our focus of analysis. 

Recent research has highlighted the correlation between the municipal share of voting for Bolsonaro and COVID-19 results [37]. However, they all relied on second-round election data and did not use placebo tests for other respiratory diseases to leverage the regression results powerfully. Without controlling for the effects of traditional right-wing votes and using first-round election data, those studies cannot separate the effects of Bolsonaro’s political pressure to reduce compliance with social distancing measures from the ecological correlation between right-wing votes and worse COVID-19 results. We expect that the effects of populism could be stronger for municipalities in the first round of the elections. In the second round, electors decide more based on the rejection of the other candidate than on the voters' adherence to the chosen one's principles and values.

As the president did not affiliate with any party, theoretical party connection should be used with caution when analyzing the effects of right-wing populism in Brazil. Hence, our methodological approach separating the effects of Bolsonarism from other ideological currents in Brazilian politics is more reliable than data mining methods.

 

 Reviewer 1

“In the introduction it is needed to present how this paper pushes forward the current knowledge in populism studies or what it brings new at international level of populism and coronavirus studies. So, the question that should be raised is how this study is positioned in the current broader CEE contexts or why the Brasilian case in this paper is so different from other cases we already know. These ideas can be then further connected in the last part of the paper and see how the findings of this paper differ to the previous ones in contexts of populism and coronavirus in different other countries.” 

“The theoretical background of the paper has to be a little bit improved with some recent studies on coronavirus and on the emergence of the far-right studies in different countries worldwide. First, I think some works on COVID-19 should be mentioned. There are several special issues launched at different journals on COVID-19. For instance the case of transnational migrant women in India is important, also the case of transnational workers from Romania who tried hard to work in western Europe (see a study in Eurasian Geography and Economics, 2020)- OK Baixada a coleçao inteira. Second, with connection to far-right populism there is the case of the AUR party in Romania – see Vesalon L. and Popescu L. in journal East European Politics (2022)”

“Moreover, I think that several examples of how the poor people are much easy to be influenced by the far-right discourse, because these people are more vulnerable. For instance, in Europe it is the case of the Roma people who are used by populists and far-right leaders as an electoral basin in order to win elections (see the study by Mereine Berki B. et al, 2020 in Geographica Pannonica ok, on how stigma against the poor Roma in Hungary can appear basing among others on historical and political manipulation of these people). The same is highlighted in an East-Central European border rural area (Romanian-Serbian) for the Roma in Romania (see a study of Covaci and Jucu, 2021 in journal Identities)”.

RESPONSE: Brazil is an interesting case to study the consequences of populism in the pandemic for many reasons highlighted at the end of the [INTRODUCTION] and [POPULISM AD COVID-19] sections. To develop this element in our text, we deepen the theoretical discussion, incorporating the reviewer's suggestions, and better justify the Brazilian case's choice by putting the dimensions of Brazilian radical right populism in perspective compared to the CEE context.

First, the populist Bolsonarist movement was made without the support of any party organization. Besides, during much of the pandemic, the president was not affiliated with any party. Second, Bolsonarism represents the most significant contemporary experience of radical far-right populism in emerging democratic countries and is unique in Latin America. Several cases of populist leaders criticize political globalization and elite controls over national political processes from a radical far-right chauvinist view in the developed world. However, recent populist experiences in Latin America had been preferentially composed of a critique of left-wing neoliberal economic globalization. The Kirchners, Venezuelan Bolivarianism, and, more recently AMLO in Mexico are examples. That said, the Brazilian case has interesting specificities to be explored further. The Brazilian case has interesting specificities that shed light on some of the consequences a victory of radical right-wing populism has had in a consolidated and dynamic democracy.

Another critical consideration not directly addressed in the manuscript is that evidence from surveys and aggregate data in Brazilian public opinion research and electoral surveys indicates that the core of Bolsonarist populism is not composed of voters among the poorest. Bolsonarists are among the middle classes and socioeconomic elites.

[INTRODUCTION]

In this paper, we analyze the Brazilian case under many factors. First, Bolsonarism is a new political movement of right-wing populism, which became the most influential political force after the 2018 presidential election. In the past, PSDB (Partido Social Democrata Brasileiro), a soft center-right party, represented right-wing views in the country; however, they were replaced by the new right-wing populist movement. By assessing the variations in right-wing voting in Brazil during the last presidential elections, we can detach the effects of traditional right-wing voting from the new populist radical right movement. 

Second, recent populist experiences in Latin America were usually marked by left-wing critiques towards neoliberal economic globalization, represented in figures such as the Kirchners in Argentina, the Venezuelan Bolivarianism, and, more recently, AMLO in Mexico. In addition, it is also different from Central and East European extreme right-wing populism has xenophobic views [4–8]. 

Third, Brazil is a data-rich environment with more than 5570 local governments and 27 states. Besides that, within the same legal framework, there is enormous variation in support for populism among Brazilian municipal entities. Hence, the effects of populism can be analyzed regardless of the characteristics of the local political system. 

Furthermore, the effects of Bolsonarism cannot be reduced to party clues, as the president was not affiliated with any party during the majority of the pandemic and changed party affiliation in the last year of his mandate. 

Bolsonarism represents the most significant contemporary experience of radical right-wing populism in an emerging country. The emergence of evangelicals as a right-wing movement has been very recent in Brazil. During the past five years, they have evolved from a small, minority group to a large, hegemonic one. Although some churches have supported left-wing governments, a mix of very conservative beliefs has become the law. For now, they are one of the main supports of Bolsonaro's politics and policies. The president has embraced the conservative view regarding family, values, and a pro-USA view [9]. In addition, Bolsonaro’s original political view has always been related to right-wing political extremism. For some authors, cruelty characterizes this kind of politics. In a democratic country, that means downplaying the effects of the disease during the pandemic or shifting the blame toward other actors [10]. 

There were several cases of populist leaders criticizing political globalization and global policy recommendations from a radical far-right chauvinist view in the developed world. However, none of them openly supported the herd immunity approach to combat the pandemic, ignoring efforts for rapid vaccine production [11]. 

Lastly, Brazil has a robust and advanced public health system, making it a particularly relevant case among emerging countries as it has more developed state health capacities than others with the same per capita income. With that being said, the Brazilian case has interesting specificities to be explored that can help shed light on some of the consequences a victory of radical right-wing populism has had in a consolidated and dynamic democracy.

[POPULISM AND COVID-19]

Features of the Brazilian context highlight what some studies emphasized [30]. Understanding new populism requires analyzing how parties enter and navigate the electoral and party systems and the content of their rhetorical appeals to the public. Bolsonaro's behavior relates to a style of politics based on bad manners, which focuses on delivering performance against political correctness [31]. 

As extensively demonstrated in the conceptual literature on populism, even if there is an ideological aspect in its constitution, it is also necessary to understand the dichotomy between the elite and the masses as a political strategy [32,33]. This discourse, or even logic of action, is less dependent on the leader and more associated with representational deficits [34–36].

(…)

As the president did not affiliate with any party, theoretical party connection should be used with caution when analyzing the effects of right-wing populism in Brazil. Hence, our methodological approach separating the effects of Bolsonarism from other ideological currents in Brazilian politics is more reliable than data mining methods.

Besides party affiliation, the Brazilian case is also interesting because of its experience with the xenophobic radical right in Latin America. Even though Brazil does not have a migration problem, only a specific issue in the state of Roraima due to the Venezuelan humanitarian crisis, the immigration problem is routinely mobilized by the populist right-wing leadership [38]. The Latin American continent has been a region with a strong affinity for leftist populist governments, with the Bolivarian and Kirchnerist experiences being the most recent demonstrations [39]. The Bolsonarist experience, in turn, is more in tune with the emergence of North Atlantic populist movements that reject globalization more in its political than economic aspects, even reverberating the notions of cruelty as a political strategy [40].

In this way, Brazil shares with Central and Eastern European countries long history of racism. In Europe, racial stigma against Roma people has deep historical roots. Stigmatization remains in the collective mentality despite efforts for integration. However, Bolsonaro's vote aligns not with racial lines but with class and regional cleavages. That indicates the emergence of a new kind of right within more fluid xenophobic lines, revigorating the long Brazilian tradition of huge racial inequalities as a direct consequence of late slavery. Collective identities aid in the establishment of essential directions around whether groups are accepted or rejected. As a result, survey evidence and aggregate data eventually indicate that the core of Bolsonarism is not composed of voters among the poorest but among the middle classes and socioeconomic elites [41–43].

“Methods and data interpretation are strong. Just to highlight in 1-2 sentences the limitations of the data and methods.”.

RESPONSE: We create a new section to discuss the limits of the study [LIMITATIONS OF THE STUDY]

Our analysis relies on ecological data. We have the leverage to estimate the effect of right-wing populism on COVID-19 results due to denialism and political pressures from the Brazilian president. However, further research should test the micro-foundations of our hypothesis using individual-level data. A preliminary analysis of Brazilian data corroborates our findings. 

Although our placebo strategy helps shed light on the relationship between COVID-19 results and political variables, our estimation does not rely on exogenous variation to estimate Bolsonarism’s effect on deaths and cases. Hence, we are confident that we can capture the structure of the relations. However, more research should be done to estimate the actual effects of Bolsonaro's politics on the number of cases and deaths caused by COVID-19 in Brazil.

A further line of inquiry is to identify how socioeconomic status modulates the effect of Bolsonarism, as the right-wing populist movement in Brazil is associated with middle- and high-income individuals.

“Because there is not a clear-cut discussion section in this paper, I would like to see (before the conclusions section) about 1-2 paragraphs connecting better the results of this study with the literature review on coronavirus and far-right populism. Also, some policy recommendations can be shortly addressed.”

RESPONSE: Very important commentary. We tried to partly answer it in the response for the 1st commentary of R1. Besides, we incorporate policy recommendations at the end of the DISCUSSION section.

We see the Brazilian case as an example of Punt Politics when national governments defer or deflect - de jure or de facto - responsibility to sub-national entities for crucial decisions that require centralized stewardship – in Brazil driven by the presidency. That resulted in fragmented, uncoordinated responses at odds with health needs and consistency with evidence. 

Results indicate that better public policy response demands more accountability than there is in Brazil. Clear responsibilities can decrease the room for populist views. That, for example, would also reduce how populists can blame others for bad public policy results. More social policies could also reduce the space populist politicians use for social cleavages to manipulate people.

“Conclusions should be a bit expanded by better showing the international, regional (Latin American) and Brasilian implications of this study.”.

RESPONSE: Very important commentary. We tried to answer it in the response for the 1st commentary of R1.

[POPULISM AND COVID-19]

In this way, Brazil shares with Central and Eastern European countries long history of racism. In Europe, racial stigma against Roma people has deep historical roots. Stigmatization remains in the collective mentality despite efforts for integration. However, Bolsonaro's vote aligns not with racial lines but with class and regional cleavages. That indicates the emergence of a new kind of right within more fluid xenophobic lines, revigorating the long Brazilian tradition of huge racial inequalities as a direct consequence of late slavery. Collective identities aid in the establishment of essential directions around whether groups are accepted or rejected. As a result, survey evidence and aggregate data eventually indicate that the core of Bolsonarism is not composed of voters among the poorest but among the middle classes and socioeconomic elites [41–43].

“The reference list is short, it is made up of about 20 references. I think the reference list should be made of about 50 sources. This aspect is in connection to my above point 1) where I suggested some references, but besides those examples authors have to add more references on extremism/nationalism/far-right populists in Europe, Latin America, Asia etc.”.

RESPONSE: Very important commentary. We incorporate a deeper theoretical discussion to justify the choice of the case, as well as the theoretical implications of our findings. 

 Reviewer 2

“As a first comment, we can consider that the theme of the article is relevant, but there are already published articles showing that there were more cases of COVID-19 and higher mortality in Brazilian municipalities where JB had more votes. The article by Xavier et al. (2022), published in The Lancet Regional Health-Americas, was widely commented on in the media. Cabral et al. (2021) and Constantino et al. (2021) published other articles on this topic. Surprisingly, none of these previous studies were cited in the present article. Thus, for the present article to be considered for publication, the authors need to describe this previous research and be convincing about what the present article can add to the findings of these authors. Without this effort, the present article will be seen as "more of the same", and the conclusions that JB had more votes where there were more cases and deaths from COVID-19 are already known”.

RESPONSE: We thank the Reviewer and Editor for their indication and incorporate in our analysis the results of Xavier et al. (2022) and Constantino et al. (2021) [POPULISM AND COVID-19].

Despite the indication of the text by Cabral et al. (2021), we chose not to incorporate it in our discussion because it is not yet published, which would violate the journal's recommendation in https://journals.plos.org/plosone/s/submission-guidelines#loc-references

[Do not cite the following sources in the reference list: Unavailable and unpublished work, including manuscripts that have been submitted but not yet accepted (e.g., “unpublished work,” “data not shown”). Instead, include those data as supplementary material or deposit the data in a publicly available database.]

Besides, we incorporate the paper by Constantino et al. (2021). in another section because it is more directly related to the electoral consequences of COVID-19 in the 2020 Brazilian municipal election than to the analysis of the mediating effects of policy on pandemic outcomes, which is our focus of analysis. 

Recent research has highlighted the correlation between the municipal share of voting for Bolsonaro and COVID-19 results [37]. However, they all relied on second-round election data and did not use placebo tests for other respiratory diseases to leverage the regression results powerfully. Without controlling for the effects of traditional right-wing votes and using first-round election data, those studies cannot separate the effects of Bolsonaro’s political pressure to reduce compliance with social distancing measures from the ecological correlation between right-wing votes and worse COVID-19 results. We expect that the effects of populism could be stronger for municipalities in the first round of the elections. In the second round, electors decide more based on the rejection of the other candidate than on the voters' adherence to the chosen one's principles and values.

As the president did not affiliate with any party, theoretical party connection should be used with caution when analyzing the effects of right-wing populism in Brazil. Hence, our methodological approach separating the effects of Bolsonarism from other ideological currents in Brazilian politics is more reliable than data mining methods.

“Some editing for English language is required throughout the manuscript due to too many mistakes. These are just a few examples found in the abstract:”.“Change “Placebo tests indicates that” to “Placebo tests indicate that”. Change "indicator for a poor response" to "indicator of a poor response". Other weird sentences and word choices need revision”. 

RESPONSE: Our sincere thanks for the indications that all suggestions were incorporated. We made a complete revision to the final version of the manuscript.

“The abstract should be self-explanatory, that is, the reader should be able to grasp the subject without referring to the main text. In this sense, it is not clear what "placebo tests" means in the abstract.”

RESPONSE: Our sincere thanks for the indication. We incorporate the suggestion in the ABSTRACT.

Placebo tests using traditional right-wing vote and data on Severe Acute Respiratory Syndrome (SARS) before the pandemic outbreak indicate that the former does not correlate with health outcomes, and the populist share of the vote does not correlate with the latter. Hence, we find strong indications that right-wing populism is connected with a poor response to the disease.

“Abstract, third line. I would suggest to use “effects of the disease” or “effects of the COVID-19” rather than “effects of SARS-CoV-2”. SARS-CoV-2 refers to the virus, while COVID-19 refers to the disease that it causes in humans”. 

RESPONSE: Our sincere thanks for the indications that all suggestions were incorporated. We made a complete revision to the final version of the manuscript.

“Introduction. It is not clear how important this sentence is in the context of the introduction: “In the past, right-wing views were represented by PSDB (Partido Social Democrata Brasileiro), a soft right-center party”. Perhaps the PSDB was closer to Tony Blair's "Third Way" vision for Europe than a far-right party, but I believe that this information is not important in the introduction of the article.”

RESPONSE: The fact that the PSDB is a traditional party of the Brazilian political right helps us to disentangle the effect of the new populist movement of the radical right, whose electorate is different, although correlated, from the previous party that got most of the votes from the right. We do not aim to evaluate the PSDB's program comparatively. However, we agree with the reviewer that the PSDB has positions more similar to Tony Blair's Third Way, its leader being part of the movement, than other right-wing parties in the Americas or Brazil. In the INTRODUCTION, we highlight that:

In this paper, we analyze the Brazilian case under many factors. First, Bolsonarism is a new political movement of right-wing populism, which became the most influential political force after the 2018 presidential election. In the past, PSDB (Partido Social Democrata Brasileiro), a soft center-right party, represented right-wing views in the country; however, they were replaced by the new right-wing populist movement. By assessing the variations in right-wing voting in Brazil during the last presidential elections, we can detach the effects of traditional right-wing voting from the new populist radical right movement. 

 

“Introduction. The authors state that “… the effects of Bolsonarism cannot be reduced to party clues, as the president was not affiliated to any party during most part of the pandemics and changed party affiliation in the last year of his mandate”. Understanding the support that JB has from the Brazilian population requires knowledge of many factors, including religious, cultural, and social issues. Brazilian society is patriarchal and conservative, and the support for JB's ideas goes far beyond party-political issues. I believe that these articles can help the authors better structure the introduction of the article by describing the social structure of the country during the pandemic period: Farias, D. B. L., Casarões, G., & Magalhães, D. (2022). Radical right populism and the politics of cruelty: The case of COVID-19 in Brazil under President Bolsonaro. Global Studies Quarterly, 2(2). Burity, J. (2021). The Brazilian conservative wave, the Bolsonaro administration, and religious actors. Brazilian Political Science Review, 15.Barberia, L. G., & Gómez, E. J. (2020). Political and institutional perils of Brazil's COVID-19 crisis. Lancet (London, England), 396(10248), 367”

RESPONSE: Our sincere thanks for the indications all suggestions were incorporated and a revision was made in INTRODUCTION the final version of the manuscript.

Bolsonarism represents the most significant contemporary experience of radical right-wing populism in an emerging country. The emergence of evangelicals as a right-wing movement has been very recent in Brazil. During the past five years, they have evolved from a small, minority group to a large, hegemonic one. Although some churches have supported left-wing governments, a mix of very conservative beliefs has become the law. For now, they are one of the main supports of Bolsonaro's politics and policies. The president has embraced the conservative view regarding family, values, and a pro-USA view [9]. In addition, Bolsonaro’s original political view has always been related to right-wing political extremism. For some authors, cruelty characterizes this kind of politics. In a democratic country, that means downplaying the effects of the disease during the pandemic or shifting the blame toward other actors [10].

“Introduction. “In short, the Brazilian president disseminated the idea that COVID-19 was just a “little flu” and combating the economic crisis was more important”. I would suggest to cite this article:Dyer, O. (2020). Covid-19: Bolsonaro under fire as Brazil hides figures. BMJ, 369, m2296”

Our sincere thanks for the indication. We incorporate the suggestion of citation.

“Introduction. In summary, the introduction section needs to be extensively reformulated in an effort to present previous articles on the topic, possible "gaps" in this research, and essential information for the reader to understand the context of the JB government's political landscape of misinformation and omission.

RESPONSE: The reviewer is entirely correct in his point, so we made a profound revision of the INTRODUCTION, deepening the contextualization of the Brazilian case, as well as better justifying the choice of the case and the relevance of the analysis of Brazil to understand the effects of radical right populism in the contemporary world.

“The methods section does not make clear how respiratory disease data were included in the statistical model to test for a "placebo effect." The methods section should contain enough information to ensure that readers can reproduce the statistical analyses”.

RESPONSE: We have reviewed the methods section to make it easier for the reader to understand the approach used in the DATA AND METHODS section.

We drew on an original dataset covering Brazil’s 5,570 municipalities to evaluate the connection between right-wing extremism and the spread of the pandemic. Our data represents one of the most extensive balanced datasets in the world, aligning key political aspects, economic infrastructure, social distancing, and health capacity with deaths and cases at a local level. As we are dealing with a count date, where the unconditional mean of the outcome is much different from its variance, we estimate negative binomial models for panel and cross-sectional data [45,46]. We control the spatial dimension using the distance between the state capital and the Brazilian federal capital and dummies for regions. Negative binomial models are adequate to handle count data as deaths and COVID-19 cases. In the appendix, as a further robustness analysis, we tested the model with cases and deaths per capita as the dependent variables, even considering that all our models control for population size.

(…)

As robustness checks for the effect of populism, we tested the impact of the moderate right-wing vote, using data from the 2014 election and data for other severe respiratory diseases in 2018 as a placebo test for overall respiratory diseases unrelated to COVID-19. Data on Severe Acute Respiratory Syndrome (SARS) was initially collected in Brazil as a response to the Zika virus pandemic. We collected data on SARS from 2018 in all Brazilian municipalities on the Health Ministry database. We assume that there are no changes concerning SARS transmission in Brazil between 2018 and 2020 regarding socioeconomic and epidemiological variables, except for the political context and the surge of COVID-19, which makes it a valuable placebo. 

Finally, we used the moderator-mediator variable distinction approach and other methods of exploring causality to explore the possible mechanism: the populist effect on social isolation compliance [47,48]. 

“ In the conclusion section, the authors argue that the "Findings show that the impact of extreme right-wing views on the pandemic has been striking in Brazil". However, in the introduction of the article, the authors discuss that "In the Brazilian case, however, party connections are less relevant, as the president was not a member of any party throughout the first and second wave of the pandemic". I think that these sentences are contradictory in some ways. It does not appear that right-wing views are related to the number of COVID-19 cases or deaths, but these relationships are determined by a wide range of issues, including traditional, conservative, religious views, and even a sense of Brazilian Mccarthyism that demonizes perceived "left-wing" social actions. So the problem does not seem to be that JB is a right-wing politician, but what causes greater numbers of the disease is his populism and conservative views of the population”.

RESPONSE: Despite not being a political party member, extensive literature shows that Bolsonaro is undeniably an extreme right-wing leader. We expect that some part of the Brazilian population has conservative views. However, the views that Bolsonaro carries are extremist ones. As a result, conservative views and populism are essential to understanding the disease's dynamic. Nevertheless, it is also true that the type of views shared (i.e., antiscience) by him and the role Bolsonaro played as president (i.e., blaming the left) makes radical right-wing politics crucial to what happened in Brazil.

“I have doubt that the statistical methods used by the authors are the most appropriate for the data. The authors do not comment on model assumptions, such as the linearity of the relationship between independent and dependent variables, and the diagnosis of homoscedasticity and distribution of residuals. The model does not include an offset variable or a population size weighting strategy. The model also does not include spatial structures that would allow testing on maps for clusters where JB has more votes and COVID-19 cases were more frequent. The previous papers by Xavier et al. (2022), Cabral et al. (2021), and Constantino et al. (2021) show maps and graphs, which are more understandable than the results shown in the present study”.

RESPONSE: We sincerely thank the reviewer for these issues. We rewrote the [DATA AND METHODS] and the [RESULTS] to make it clear to the reader how we address all these questions. Regarding the model assumptions, we employ count data models that better fit the dependent variables when they are count variables. Negative binomial models are adequate to handle count data as deaths and covid cases. However, we also included an analysis using offset variables. We estimated all complete models using cases of SARS per 1000 inhabitants, Covid per 1000 inhabitant cases, and Covid per 1000 inhabitant deaths as the dependent variables. The findings remain the same. We show the complete results in the Appendix. 

Concerning standard errors, we estimated them clustered by states. 

For the spatial dimension, we modeled it with the following specification f(.), always using three variables for each municipality.

f(.)=φ+φ_1 distance to the federal capital+φ_2 distance to the state capital+dummies for regions

We estimated all complete models with this specification. We believe it is enough to handle the spatial spread of the disease and the spatial distribution of the share of Bolsonaro's vote.

Finally, we inserted [GRAPHIC 01 – RESPIRATORY DISEASES IN BRAZIL] and [GRAPHIC 02 – THE RIGHT-WING VOTE IN BRAZIL] to make the paper more easily for readers. 

[DATA AND METHODS] [RESULTS]

As we are dealing with a count date, where the unconditional mean of the outcome is much different from its variance, we estimate negative binomial models for panel and cross-sectional data [45,46]. We control the spatial dimension using the distance between the state capital and the Brazilian federal capital and dummies for regions. Negative binomial models are adequate to handle count data as deaths and COVID-19 cases. In the appendix, as a further robustness analysis, we tested the model with cases and deaths per capita as the dependent variable, even considering that all our models control for population size.

(…)

The findings are the same when we use cases and deaths of Covid per 1000 inhabitant as the dependent variable. Table 4A in the appendix presents the results.

(…)

The results are the same when we use cases of SARS per 1000 inhabitants, Covid per 1000 inhabitant cases, and Covid per 1000 inhabitant deaths as the dependent variables. Table 4B in the appendix presents the results.

“Potential limitations of the study are not presented.”

We create a new section to discuss the limits of the study [LIMITATIONS OF THE STUDY]

Our analysis relies on ecological data. We have the leverage to estimate the effect of right-wing populism on COVID-19 results due to denialism and political pressures from the Brazilian president. However, further research should test the micro-foundations of our hypothesis using individual-level data. A preliminary analysis of Brazilian data corroborates our findings. 

Although our placebo strategy helps shed light on the relationship between COVID-19 results and political variables, our estimation does not rely on exogenous variation to estimate Bolsonarism’s effect on deaths and cases. Hence, we are confident that we can capture the structure of the relations. However, more research should be done to estimate the actual effects of Bolsonaro's politics on the number of cases and deaths caused by COVID-19 in Brazil.

A further line of inquiry is to identify how socioeconomic status modulates the effect of Bolsonarism, as the right-wing populist movement in Brazil is associated with middle- and high-income individuals.

“ The discussion section is very shallow and it needs major modification.”

RESPONSE: We have reviewed the DISCUSSION section considering all suggestions of the Reviewers.

“At the end of the article, the sentence beginning with “Lastly, an important research topic on the effects of populism…” needs some revision, and this argument could be supported by the following articles: Daniels, J. P. (2021). Health experts slam Bolsonaro's vaccine comments. The Lancet, 397(10272), 361. Boschiero, M. N., Palamim, C. V. C., & Marson, F. A. L. (2021). COVID-19 vaccination on Brazil and the crocodile side-effect. Ethics, medicine, and public health, 17, 100654”. 

Our sincere thanks for the indication. We incorporate the suggestion..

 

 Reviewer 3

“I have several recommendations that I think will improve the overall contribution, which I already find compelling. First, I suggest adding a “limitations” section along the lines of what we’ve seen recently in many public health and medical journal publications surrounding covid-19. Most of these limitations are methodological, which I think is completely acceptable given the real-time data collection and analysis in question.”.

RESPONSE: We create a new section to discuss the limits of the study [LIMITATIONS OF THE STUDY]

Our analysis relies on ecological data. We have the leverage to estimate the effect of right-wing populism on COVID-19 results due to denialism and political pressures from the Brazilian president. However, further research should test the micro-foundations of our hypothesis using individual-level data. A preliminary analysis of Brazilian data corroborates our findings. 

Although our placebo strategy helps shed light on the relationship between COVID-19 results and political variables, our estimation does not rely on exogenous variation to estimate Bolsonarism’s effect on deaths and cases. Hence, we are confident that we can capture the structure of the relations. However, more research should be done to estimate the actual effects of Bolsonaro's politics on the number of cases and deaths caused by COVID-19 in Brazil.

A further line of inquiry is to identify how socioeconomic status modulates the effect of Bolsonarism, as the right-wing populist movement in Brazil is associated with middle- and high-income individuals.

“causal identification is very difficult in this context, which I think is reasonable and currently common in the top medical journals for work on covid. However, I suggest moderating some of the claims in the paper and describing why the models presented represent the best-case scenario in the current climate and how the results can still help build theory and inform government practice. Another limitation is simply the lack of municipal level covariates available and/or the way that aggregating to the municipal level obscures unobserved and (probably) unobservable submunicipal variation.”

RESPONSE: Although our placebo strategy help to shed light on the relations between covid19 results and political variables, our estimation does not rely on exogenous variation to estimate the effect of Bolsonarism on deaths and cases. Hence, we are confident that we can capture the structure of the relations. However, more research should be done to more reliably estimate the actual effects of Bolsonaro's politics on the number of cases and deaths by COVID-19 in Brazil.

“I suggest defending the case selection a little more thoroughly. For example, are the results from Brazil generalizable beyond the country? Why/why not? I agree with the authors’ rationale and I think they should make an even larger claim: that brazil, due to data availability and municipal variation, is the only country where they could plausibly test hypotheses against such broad, deep data.” 

RESPONSE: We incorporate a deeper discussion of the case selection and the leverage that studying of Brazilian case has in the literature both in the INTRODUCTION and in the POPULISM AND COVID-19 sections.

[INTRODUCTION]

In this paper, we analyze the Brazilian case under many factors. First, Bolsonarism is a new political movement of right-wing populism, which became the most influential political force after the 2018 presidential election. In the past, PSDB (Partido Social Democrata Brasileiro), a soft center-right party, represented right-wing views in the country; however, they were replaced by the new right-wing populist movement. By assessing the variations in right-wing voting in Brazil during the last presidential elections, we can detach the effects of traditional right-wing voting from the new populist radical right movement. 

Second, recent populist experiences in Latin America were usually marked by left-wing critiques towards neoliberal economic globalization, represented in figures such as the Kirchners in Argentina, the Venezuelan Bolivarianism, and, more recently, AMLO in Mexico. In addition, it is also different from Central and East European extreme right-wing populism has xenophobic views [4–8]. 

Third, Brazil is a data-rich environment with more than 5570 local governments and 27 states. Besides that, within the same legal framework, there is enormous variation in support for populism among Brazilian municipal entities. Hence, the effects of populism can be analyzed regardless of the characteristics of the local political system. 

Furthermore, the effects of Bolsonarism cannot be reduced to party clues, as the president was not affiliated with any party during the majority of the pandemic and changed party affiliation in the last year of his mandate. 

Bolsonarism represents the most significant contemporary experience of radical right-wing populism in an emerging country. The emergence of evangelicals as a right-wing movement has been very recent in Brazil. During the past five years, they have evolved from a small, minority group to a large, hegemonic one. Although some churches have supported left-wing governments, a mix of very conservative beliefs has become the law. For now, they are one of the main supports of Bolsonaro's politics and policies. The president has embraced the conservative view regarding family, values, and a pro-USA view [9]. In addition, Bolsonaro’s original political view has always been related to right-wing political extremism. For some authors, cruelty characterizes this kind of politics. In a democratic country, that means downplaying the effects of the disease during the pandemic or shifting the blame toward other actors [10]. 

There were several cases of populist leaders criticizing political globalization and global policy recommendations from a radical far-right chauvinist view in the developed world. However, none of them openly supported the herd immunity approach to combat the pandemic, ignoring efforts for rapid vaccine production [11]. 

Lastly, Brazil has a robust and advanced public health system, making it a particularly relevant case among emerging countries as it has more developed state health capacities than others with the same per capita income. With that being said, the Brazilian case has interesting specificities to be explored that can help shed light on some of the consequences a victory of radical right-wing populism has had in a consolidated and dynamic democracy.

[POPULISM AND COVID-19]

Features of the Brazilian context highlight what some studies emphasized [30]. Understanding new populism requires analyzing how parties enter and navigate the electoral and party systems and the content of their rhetorical appeals to the public. Bolsonaro's behavior relates to a style of politics based on bad manners, which focuses on delivering performance against political correctness [31]. 

As extensively demonstrated in the conceptual literature on populism, even if there is an ideological aspect in its constitution, it is also necessary to understand the dichotomy between the elite and the masses as a political strategy [32,33]. This discourse, or even logic of action, is less dependent on the leader and more associated with representational deficits [34–36].

(…)

As the president did not affiliate with any party, theoretical party connection should be used with caution when analyzing the effects of right-wing populism in Brazil. Hence, our methodological approach separating the effects of Bolsonarism from other ideological currents in Brazilian politics is more reliable than data mining methods.

Besides party affiliation, the Brazilian case is also interesting because of its experience with the xenophobic radical right in Latin America. Even though Brazil does not have a migration problem, only a specific issue in the state of Roraima due to the Venezuelan humanitarian crisis, the immigration problem is routinely mobilized by the populist right-wing leadership [38]. The Latin American continent has been a region with a strong affinity for leftist populist governments, with the Bolivarian and Kirchnerist experiences being the most recent demonstrations [39]. The Bolsonarist experience, in turn, is more in tune with the emergence of North Atlantic populist movements that reject globalization more in its political than economic aspects, even reverberating the notions of cruelty as a political strategy [40].

“Defending the particular timeframe under consideration should also be part of the next revision. I see lots of reasons to focus on the timeframe under consideration in the article, but I would like to see those reasons articulated thoroughly. For example, because the covid response shifted from NPIs to vaccines after 2020.”

RESPONSE: In the Data and Methods section, we explicitly explain why we use 2020 as our particular timeframe. 

Our time frame concerns only 2020 since a massive vaccination campaign started in January 2021, despite the difficulties created by the federal government for a broad vaccination of the Brazilian population. Consequently, the Covid response involves more strategies than NPIs. Future studies should analyze the implications.

“Finally, there is recent literature on Brazil and Mexico and on subnational covid issues in Latin America in general that should be included in the review and with which this article can enter into conversation: 

Touchton, Michael, Felicia Marie Knaul, Héctor Arreola-Ornelas, Thalia Porteny, Mariano Sánchez, Oscar Méndez, Marco Faganello et al. "A partisan pandemic: state government public health policies to combat COVID-19 in Brazil." BMJ global health 6, no. 6 (2021): e005223. Knaul, F. M., Touchton, M. M., Arreola-Ornelas, H., Calderon-Anyosa, R., Otero-Bahamón, S., Hummel, C., ... & Sanchez-Talanquer, M. (2022). Strengthening Health Systems To Face Pandemics: Subnational Policy Responses To COVID-19 In Latin America: Study examines policy responses to COVID-19 in Latin America. Health Affairs, 41(3), 454-462. Testa, Paul F., Richard Snyder, Eva Rios, Eduardo Moncada, Agustina Giraudy, and Cyril Bennouna. "Who Stays at Home? The Politics of Social Distancing in Brazil, Mexico, and the United States during the COVID-19 Pandemic." Journal of Health Politics, Policy and Law (2021). Castro, Marcia C., Sun Kim, Lorena Barberia, Ana Freitas Ribeiro, Susie Gurzenda, Karina Braga Ribeiro, Erin Abbott, Jeffrey Blossom, Beatriz Rache, and Burton H. Singer. "Spatiotemporal pattern of COVID-19 spread in Brazil." Science 372, no. 6544 (2021): 821-826. Knaul, Felicia, Héctor Arreola-Ornelas, Thalia Porteny, Michael Touchton, Mariano Sánchez-Talanquer, Óscar Méndez, Salomón Chertorivski et al. "Not far enough: Public health policies to combat COVID-19 in Mexico’s states." Plos one 16, no. 6 (2021): e0251722.”

RESPONSE: Our sincere thanks for the indication. We incorporated all the suggestions.

---

## [Decision Letter · Decision Letter 1]

25 Nov 2022

Populism and health. An evaluation of the effects of right-wing populism on the COVID-19 pandemic in Brazil

PONE-D-22-14260R1

Dear Dr. Fernandes,

We’re pleased to inform you that your manuscript has been judged scientifically suitable for publication and will be formally accepted for publication once it meets all outstanding technical requirements.

Kind regards,

Diego Augusto Santos Silva, Ph.D.

Academic Editor

PLOS ONE

**Comments to the Author**

1. If the authors have adequately addressed your comments raised in a previous round of review and you feel that this manuscript is now acceptable for publication, you may indicate that here to bypass the “Comments to the Author” section, enter your conflict of interest statement in the “Confidential to Editor” section, and submit your "Accept" recommendation.

Reviewer #1: All comments have been addressed

Reviewer #3: All comments have been addressed

2. Is the manuscript technically sound, and do the data support the conclusions?

Reviewer #1: Yes

Reviewer #3: Yes

3. Has the statistical analysis been performed appropriately and rigorously? 

Reviewer #1: Yes

Reviewer #3: Yes

4. Have the authors made all data underlying the findings in their manuscript fully available?

Reviewer #1: Yes

Reviewer #3: Yes

5. Is the manuscript presented in an intelligible fashion and written in standard English?

Reviewer #1: Yes

Reviewer #3: Yes

6. Review Comments to the Author

Reviewer #1: (No Response)

Reviewer #3: I am satisfied with the revised submission- all comments have been addressed and I recommend acceptance for publication

7. PLOS authors have the option to publish the peer review history of their article (what does this mean?). If published, this will include your full peer review and any attached files.

Reviewer #1: No

Reviewer #3: No

---

## [Editor Report · Acceptance letter]

2 Dec 2022

PONE-D-22-14260R1 

Populism and health. An evaluation of the effects of right-wing populism on the COVID-19 pandemic in Brazil 

Dear Dr. Fernandes:

I'm pleased to inform you that your manuscript has been deemed suitable for publication in PLOS ONE. Congratulations! Your manuscript is now with our production department. 

Kind regards, 

on behalf of

Dr. Diego Augusto Santos Silva 

Academic Editor

PLOS ONE